# Isometric 3D Adversarial Examples in the Physical World

**Yibo Miao**[1,3*], **Yinpeng Dong**[2,3†], **Jun Zhu**[2,3,4,5], **Xiao-Shan Gao**[1†]

[1] KLMM, UCAS, Academy of Mathematics and Systems Science,
Chinese Academy of Sciences, Beijing 100190, China
[2] Dept. of Comp. Sci. & Tech., Institute for AI, Tsinghua-Bosch Joint ML Center,
THBI Lab, BNRist Center, Tsinghua University, Beijing 100084, China
[3] RealAI   [4] Peng Cheng Laboratory   [5] Pazhou Laboratory (Huangpu), Guangzhou, China
`yibomiao21@163.com, {dongyinpeng, dcszj}@tsinghua.edu.cn, xgao@mmrc.iss.ac.cn`

## Abstract

3D deep learning models are shown to be as vulnerable to adversarial examples as 2D models. However, existing attack methods are still far from stealthy and suffer from severe performance degradation in the physical world. Although 3D data is highly structured, it is difficult to bound the perturbations with simple metrics in the Euclidean space. In this paper, we propose a novel $\epsilon$-isometric ($\epsilon$-ISO) attack to generate natural and robust 3D adversarial examples in the physical world by considering the geometric properties of 3D objects and the invariance to physical transformations. For naturalness, we constrain the adversarial example to be $\epsilon$-isometric to the original one by adopting the Gaussian curvature as a surrogate metric guaranteed by a theoretical analysis. For invariance to physical transformations, we propose a maxima over transformation (MaxOT) method that actively searches for the most harmful transformations rather than random ones to make the generated adversarial example more robust in the physical world. Experiments on typical point cloud recognition models validate that our approach can significantly improve the attack success rate and naturalness of the generated 3D adversarial examples than the state-of-the-art attack methods.

## 1  Introduction

Deep neural networks (DNNs) have achieved unprecedented performance on numerous tasks, including 2D image classification [33, 25, 27] and 3D point cloud recognition [48, 49, 68]. However, DNNs are vulnerable to adversarial examples [59, 20] — inputs crafted by adding imperceptible perturbations to original examples that can cause misclassification of the victim model. Adversarial examples are prevalent in various domains beyond images, including texts [28], speeches [84] and 3D objects [73]. As deep 3D point cloud recognition has been adopted in safety-critical applications, such as autonomous driving [5, 85], robotics [65, 94], medical image processing [60], it has garnered increasing attention to studying the adversarial robustness of 3D point cloud recognition models [6].

However, the existing adversarial attacks on point cloud recognition models are still far from stealthy and suffer from drastic performance degeneration in the physical world. There is usually a trade-off between the stealthiness and the real-world attacking performance, making it challenging to achieve the best of both worlds. Early methods [79, 73, 42] adopt gradient-based attacks to add, remove, and modify points, but they are limited to digital-world attacks. The KNN attack [63] and the $GeoA^3$ attack [69] constrain the smoothness of the adversarial point clouds and reconstruct adversarial

---

*This work was done when Yibo Miao was intern at RealAI, Inc; †Corresponding authors.

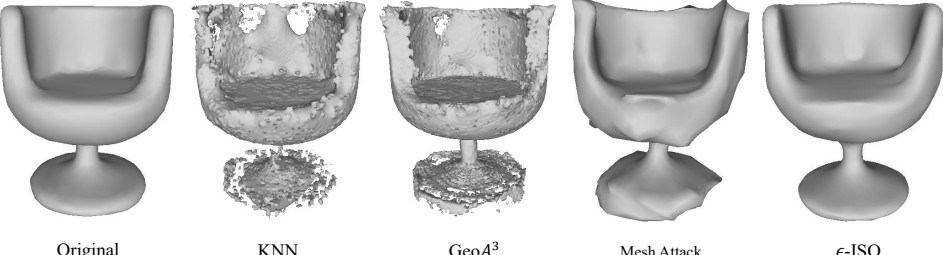

| Original | KNN | $GeoA^3$ | Mesh Attack | $\epsilon$-ISO |
|----------|-----|----------|-------------|----------------|

Figure 1: An illustration of adversarial objects crafted by KNN attack [63], $GeoA^3$ attack [69], Mesh Attack [88] and our $\epsilon$-ISO attack against the PointNet model: KNN attack and $GeoA^3$ attack can produce unnatural adversarial objects (and often low success rates); Mesh Attack can generate a lot of distortions; while $\epsilon$-ISO attack improves the naturalness of the 3D adversarial sample and ensures the consistency between the intrinsic geometric properties of the adversarial and original 3D objects [8].

meshes from the point clouds that can be 3D-printed in the physical world. Although these works demonstrate successful physical attacks, point cloud reconstruction introduces large noises and errors, resulting in low attack success rates and unnaturalness of the adversarial objects in the physical world. Mesh Attack [88] is recently proposed to perturb the mesh representation of 3D objects, which improves the success rate but often creates large distortions that can be easily detected by humans as anomalies, as illustrated in Fig. 1. Overall, it is difficult to achieve both the naturalness and effectiveness of 3D adversarial attacks in the physical world, which we think is largely due to the lack of an appropriate metric to characterize the naturalness of 3D data.

To address these issues, we propose an $\epsilon$-isometric ($\epsilon$-**ISO**) attack method to generate natural and robust 3D adversarial examples in the physical world against point cloud recognition models. The $\epsilon$-ISO attack improves the naturalness of the 3D adversarial example by constraining it to be $\epsilon$-isometric (see Definition 1) to the original one, which guarantees the consistency between the intrinsic geometric properties of two 3D objects [8]. We theoretically demonstrate that Gaussian curvature (see Definition 2) can be used to provide a sufficient condition to ensure that two surfaces are $\epsilon$-isometric. Due to the computable and differentiable nature of Gaussian curvature, we adopt it as a new regularization loss to practically generate natural 3D adversarial examples. To improve the robustness of 3D adversarial examples under physical transformations, we further propose a maxima over transformation (**MaxOT**) method that actively searches for the most harmful transformations rather than random ones [1] for optimization. Armed with Bayesian optimization that provides better initialization of the transformations, MaxOT is able to find a set of diverse worst-case transformations, leading to improved performance of the 3D adversarial examples in the physical world.

We conduct extensive experiments to evaluate the performance of our method on attacking typical point cloud recognition models [48, 49, 68]. Results demonstrate that, in comparison with the alternative state-of-the-art attack methods [63, 69, 88], $\epsilon$-ISO attack achieves higher success rates, while making the generated adversarial examples more natural and robust under physical transformations. A physical-world experiment is conducted by 3D-printing the adversarial meshes and re-scanning the objects for evaluation, which also validates the effectiveness of our method.

## 2 Related work

**Deep learning on 3D point clouds.** Deep 3D point cloud recognition [48, 21, 74, 76, 81, 50, 61] has emerged in recent years with various applications in many fields, such as 3D object classification [55, 37, 83, 86], 3D scene segmentation [22, 67, 77, 26], and 3D object detection in autonomous driving [97, 78]. One of the pioneering works is PointNet [48], which directly applies a multilayer perceptron to learn point features and aggregates them in an efficient way using a max-pool module. PointNet++ [49] and a large number of later works [13, 43, 80] are built on PointNet to further capture fine-grained local structure information from the neighborhood of each point. Recently, some works have focused on designing special convolutions on 3D domains [2, 39, 44, 62] or developing graph neural networks [18, 52, 53, 68] to improve point cloud recognition.

**3D adversarial attacks.** Following the previous studies on adversarial machine learning in the 2D image domain [59, 34, 4, 10, 19, 82, 11, 12], many works [73, 3, 41] apply adversarial attacks to the

3D point cloud domain. Xiang et al. [73] proposed point generation attacks by adding a limited number of synthetic points to the point cloud. Recently, more studies [70, 93] use gradient-based attack methods to identify key points from the point cloud for deletion. More point perturbation attacks [24, 45, 91, 9] learn to perturb the xyz coordinates of each point through a C&W framework [4] based on metrics defined in the Euclidean space. Zhao et al. [92] attack by the isometric transformations in the Euclidean space such as rotation. It is worth noting that we consider isometric mappings between surfaces, which is essentially different from [92]. Later works [35, 87] further apply iterative gradient methods to achieve more advanced adversarial perturbations. Besides, other works consider generative models [96], 3D data attacks [64, 57], adversarial robustness [89, 58], attacks against LIDAR [36, 30], autonomous driving [56, 90], backdoor attacks [38, 75], etc., in the 3D domain. However, the existing attacks on 3D point cloud recognition are still far from stealthy and the only three methods that consider the physical-world attacks [63, 69, 88] are not very effective. In this paper, we surpass the performance of previous methods by proposing a novel $\epsilon$-isometric ($\epsilon$-ISO) attack method to generate natural and robust 3D adversarial examples in the physical world.

## 3 Methodology

We now formally present $\epsilon$**-ISO attack**. We first present the general problem formulation, and then describe how $\epsilon$-ISO attack enhances the imperceptibility and robustness of the generated 3D adversarial samples, respectively.

### 3.1 Problem formulation

To generate 3D adversarial objects in the physical world, it is more straightforward to perturb the mesh representation of 3D objects rather than point clouds [88] since the reconstruction process can incur large errors [63, 69]. A mesh $\mathcal{M} = (\mathcal{V}, \mathcal{F})$ is an approximate shape representation of its underlying surface, where $\mathcal{V} := \{v_i\}_{i=1}^{n_v}$ is the set of $n_v$ vertices of $xyz$ coordinates, and $\mathcal{F} := \{z_i\}_{i=1}^{n_f}$ is the set of $n_f$ triangle faces represented by the indices of vertices. We let $S$ denote a differentiable sampling process such that $\mathcal{P} := S(\mathcal{M}) \in \mathcal{X}$ is the point cloud obtained by randomly sampling on the mesh surface, where $\mathcal{X}$ is the set of all point clouds. We let $y \in \mathcal{Y}$ denote the corresponding ground-truth label of $\mathcal{M}$ as well as $\mathcal{P}$.

In this paper, we focus on the challenging *targeted attacks* against deep 3D point cloud classification models [48, 49, 68]. Given a point cloud classifier $f : \mathcal{X} \to \mathcal{Y}$, the goal of the attack is to generate an adversarial mesh $\mathcal{M}_{adv} = (\mathcal{V}_{adv}, \mathcal{F})$ for the original one $\mathcal{M}$ with vertex perturbations such that the sampled point cloud $\mathcal{P}_{adv} := S(\mathcal{M}_{adv})$ will be misclassified by $f$ as the target class $y^*(\neq y)$. In general, the perturbation should be small to make the adversarial mesh $\mathcal{M}_{adv}$ inconspicuous under human inspection. Thus, the optimization problem of generating the adversarial mesh can be generally formulated as

$$\min_{\mathcal{M}_{adv}} \mathcal{L}_f\left(S\left(\mathcal{M}_{adv}\right), y^*\right) + \beta \cdot \mathcal{R}\left(\mathcal{M}_{adv}, \mathcal{M}\right), \tag{1}$$

where $\mathcal{L}_f$ is the loss that facilitates the misclassification of $\mathcal{P}_{adv}$ to $y^*$, $\mathcal{R}$ is the regularization term that minimizes a perceptibility distance between $\mathcal{M}_{adv}$ and $\mathcal{M}$, and $\beta$ is a balancing hyperparameter between these two losses. In this paper, we try to develop a stealthy and robust attack method by proposing a new regularization loss $\mathcal{R}$ based on Gaussian curvature with theoretical guarantees to remain the naturalness as well as a new attacking loss $\mathcal{L}_f$ to enhance the robustness of the generated 3D adversarial objects under physical transformations. $\mathcal{R}$ and $\mathcal{L}_f$ will be introduced in the following.

### 3.2 $\epsilon$-ISO attack

Most of the existing 3D adversarial attacks only consider the constraints $\mathcal{R}$ defined in the Euclidean space [15, 81, 23]. The generated adversarial examples have noticeable point outliers that cause spikes to appear on the object's surface, thus losing the naturalness. Moreover, the outliers are more easily removed and defended against. The main reason is that the existing methods do not consider the geometric properties of the 3D objects. In differential geometry, isometric mapping guarantees the consistency of the intrinsic geometric features of two objects [8]. Therefore, we propose a constraint loss $\mathcal{R}$ based on $\epsilon$-isometric mapping to restrict the naturalness of 3D adversarial objects. We first give the definition of $\epsilon$-isometric below.

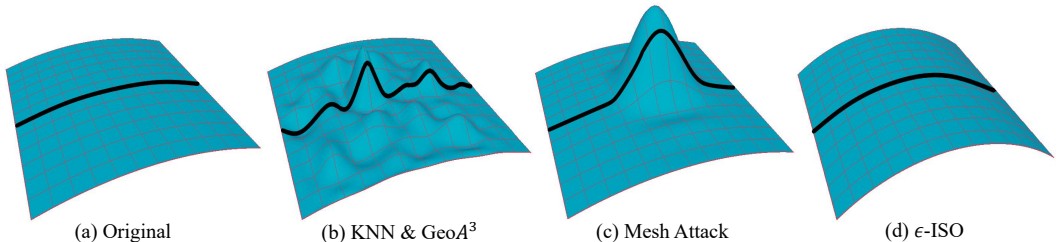

(a) Original         (b) KNN & Geo$A^3$         (c) Mesh Attack         (d) $\epsilon$-ISO

Figure 2: An illustration of $\epsilon$-isometric attack. (a): Original mesh. (b) and (c): Adversarial meshes generated by KNN & $GeoA^3$ attack and Mesh Attack, respectively. They consider only the constraints defined in the Euclidean space, and the curve lengths (shown as the black curves) of the generated adversarial examples differ significantly from those of the original samples, which do not satisfy $\epsilon$-isometric and lose naturalness. (d): Adversarial mesh generated by $\epsilon$-ISO. We consider the geometric features of 3D objects and constrain the 3D adversarial example to be $\epsilon$-isometric to the original one, such that the curve lengths of the generated adversarial samples vary little and have naturalness.

**Definition 1.** *Let $S$ and $\tilde{S}$ denote two surfaces of $\mathbb{R}^3$. A diffeomorphism $\varphi : S \to \tilde{S}$ is called an $\epsilon$-isometric mapping if there exists a constant $n$ such that it takes any local curve $C$ in $S$ to a curve $\tilde{C} = \varphi(C)$ in $\tilde{S}$ satisfying $(1 - n\epsilon)s(C) < s(\tilde{C}) < (1 + n\epsilon)s(C)$ where $s(\cdot)$ is the length. The surfaces $S$ and $\tilde{S}$ are then said to be $\epsilon$-isometric.*

As shown in Fig. 2, Fig. 2(a) is the original mesh, Fig. 2(b) is the adversarial mesh generated by KNN attack and $GeoA^3$ attack, and Fig. 2(c) is the adversarial mesh generated by Mesh Attack. These three methods only consider the constraints defined in the Euclidean space, and the curve lengths of the generated adversarial samples differ greatly from those of the original samples, which are not $\epsilon$-isometric and lose naturalness. Fig. 2(d) is the adversarial mesh generated by our proposed $\epsilon$-ISO attack. We consider the geometric features of 3D objects to generate natural adversarial examples by constraining them to be $\epsilon$-isometric to the original examples (i.e., the curve length of the resulting adversarial examples varies very little). However, it is intractable to directly optimize the adversarial mesh to be $\epsilon$-isometric as the original one. Therefore, we give the definition of the Gaussian curvature of the surface from [8].

**Definition 2.** *Let $S$ be a surface of $\mathbb{R}^3$ parameterized by $\boldsymbol{r} := \boldsymbol{r}(u, v) = [x(u, v), y(u, v), z(u, v)]$, where $(u, v) \in \mathbb{R}^2$. We let $\boldsymbol{r}_u, \boldsymbol{r}_v$ denote the partial derivatives of $\boldsymbol{r}$ w.r.t. $u$ and $v$, $\boldsymbol{r}_{uu}, \boldsymbol{r}_{uv}, \boldsymbol{r}_{vv}$ denote the second partial derivatives of $\boldsymbol{r}$, and $\wedge, \langle \cdot, \cdot \rangle$ denote the outer product and inner product, respectively. The parametrization thus defines unit normal vectors $\boldsymbol{n} := \frac{\boldsymbol{r}_u \wedge \boldsymbol{r}_v}{|\boldsymbol{r}_u \wedge \boldsymbol{r}_v|}$ of the surface $S$. We denote the eigenvalues of the coefficient matrix of the Weingarten map $\begin{bmatrix} L & M \\ M & N \end{bmatrix} \begin{bmatrix} E & F \\ F & G \end{bmatrix}^{-1}$ as $k_1$ and $k_2$, where $E = \langle \boldsymbol{r}_u, \boldsymbol{r}_u \rangle$, $F = \langle \boldsymbol{r}_u, \boldsymbol{r}_v \rangle$ and $G = \langle \boldsymbol{r}_v, \boldsymbol{r}_v \rangle$ are coefficients of the first fundamental form and $L = \langle \boldsymbol{r}_{uu}, \boldsymbol{n} \rangle$, $M = \langle \boldsymbol{r}_{uv}, \boldsymbol{n} \rangle$ and $N = \langle \boldsymbol{r}_{vv}, \boldsymbol{n} \rangle$ are coefficients of the second fundamental form. The Gaussian curvature is defined as $K = k_1 k_2 = \frac{LN - M^2}{EG - F^2}$.*

**Remark 1.** The Gaussian curvature intrinsically measures the bending degree of the surface reflected by the Gaussian mapping. Let the area element of the surface $S$ be $dA = \langle \boldsymbol{r}_u \wedge \boldsymbol{r}_v, \boldsymbol{n} \rangle \, du dv$ and the area element under the Gaussian mapping $g : \begin{matrix} S \in \mathbb{R}^3 \to S^2 \\ \boldsymbol{r}(u, v) \to \boldsymbol{n}(u, v) \end{matrix}$ be $dA' = \langle \boldsymbol{n}_u \wedge \boldsymbol{n}_v, \boldsymbol{n} \rangle \, du dv$. From $\boldsymbol{n}_u \wedge \boldsymbol{n}_v = K \boldsymbol{r}_u \wedge \boldsymbol{r}_v$ (proof in Appendix D), we obtain

$$\lim_{D \to P} \frac{\text{Area}\,(g(D))}{\text{Area}\,(D)} = \lim_{D \to P} \frac{\int_{g(D)} dA'}{\int_D dA} = \lim_{D \to P} \frac{\int_D K dA}{\int_D dA} = K(P). \tag{2}$$

Eq. (2) illustrates that the geometric meaning of Gaussian curvature is the ratio of the area of the domain at the point $P$ on the surface $S$ and the area of the domain at the corresponding point under the Gaussian mapping, i.e., the bending degree of the surface reflected by the Gaussian mapping.

Based on Definitions 1 and 2, we have the following theorem.

**Theorem 1** (proof in Appendix A). *Let $S$ and $\tilde{S}$ denote two surfaces of $\mathbb{R}^3$; $\varphi : S \to \tilde{S}$ denote a diffeomorphism that takes a point $v$ in $S$ to point $v' = \varphi(v)$ in $\tilde{S}$; and $K(\cdot)$ be the Gaussian curvature of the points. If $|K(v) - K(v')| < \epsilon$ for any point $v$, then the surfaces $S$ and $\tilde{S}$ are $\epsilon$-isometric.*

Theorem 1 indicates that to make two surfaces $\epsilon$-isometric, one can constrain their Gaussian curvatures. Since the Gaussian curvature is computable and differentiable w.r.t. vertices, we adopt it to constrain the naturalness of 3D adversarial meshes as

$$\mathcal{R}_{Gauss}\left(\mathcal{M}_{adv}, \mathcal{M}\right) = \frac{1}{n_v} \sum_{v \in \mathcal{V}, v' = \varphi(v) \in \mathcal{V}_{adv}} \left\| K\left(v'\right) - K(v) \right\|_2^2, \tag{3}$$

where $\varphi(\cdot)$ is the corresponding mapping between vertices in $\mathcal{V}$ and $\mathcal{V}_{adv}$. We follow the Gauss-Bonnet formula [7] to calculate the Gaussian curvature of the vertices as

$$K(v) = \frac{2\pi - \sum_{i \in N(v)} \theta_i(v)}{A(v)}, \tag{4}$$

where $A(\cdot)$ is the area of the vertex neighborhood, i.e., the area of the polygonal region joined by the consecutive midpoints of triangles incident on the vertex of interest, $N(v)$ is the set of faces containing $v$, and $\theta_i(v)$ is the interior angle of the face at vertex $v$. Note that the value of $\sum_{i \in N(v)} \theta_i(v)$ for a plane is $2\pi$ and the Gaussian curvature is $0$. The more curved the surface, the smaller the value of $\sum_{i \in N(v)} \theta_i(v)$ and the larger the Gaussian curvature.

In addition, we prevent the generated adversarial meshes from self-intersecting by using the Laplace loss [16], denoted as $\mathcal{R}_{Lap}\left(\mathcal{M}_{adv}\right)$, which represents the distance between a vertex and its nearest neighbor's center of mass, and the edge length loss [66], denoted as $\mathcal{R}_{edge}\left(\mathcal{M}_{adv}\right)$, which represents the smoothness of the surface. Thus, the overall regularization term can be expressed as:

$$\mathcal{R}\left(\mathcal{M}_{adv}, \mathcal{M}\right) = \lambda_1 \cdot \mathcal{R}_{Gauss}\left(\mathcal{M}_{adv}, \mathcal{M}\right) + \lambda_2 \cdot \mathcal{R}_{Lap}\left(\mathcal{M}_{adv}\right) + \lambda_3 \cdot \mathcal{R}_{edge}\left(\mathcal{M}_{adv}\right), \tag{5}$$

where $\lambda_1$, $\lambda_2$ and $\lambda_3$ are balancing hyperparameters.

### 3.3 Improving the robustness under physical transformations

Besides concerning the naturalness of 3D adversarial examples, we further enhance their robustness under physical transformations, such as 3D rotations, affine projections, cutouts, etc. A common method is the expectation over transformation (EOT) algorithm [1], which optimizes the adversarial example over the distribution of different transformations. However, it is still challenging to maintain the attacking performance under various physical transformations. As shown in the experiments, after using the EOT algorithm, there are still some transformations that the generated adversarial examples are not robust to, leading to a reduction of the attack success rate.

To address this issue, our key insight is to consider the worst-case transformations rather than their expectation, since if the adversarial examples are resistant to the most harmful physical transformations, they can also resist much weaker transformations, inspired by adversarial training [46]. Therefore, we propose a **maxima over transformation (MaxOT)** algorithm to actively search for physical transformations that maximize the misclassification loss. The loss function $\mathcal{L}_f$ is thus formulated as:

$$\mathcal{L}_f(S(\mathcal{M}_{adv}), y^*) = \max_{T^* \subset T} \mathbb{E}_{t \in T^*} \mathcal{L}_{ce}\left(t(S(\mathcal{M}_{adv})), y^*\right), \tag{6}$$

where $T$ contains all possible transformations, $T^*$ is a subset of transformations in $T$, and $\mathcal{L}_{ce}$ is the cross-entropy loss. Note that in Eq. (6) we consider a subset of transformations $T^*$ rather than a single one because the loss landscape w.r.t. transformations is largely non-convex and contains many local maxima [14]. Thus we aim to find a set of diverse worst-case transformations. By integrating Eq. (6) into Eq. (1), it forms a minimax optimization problem, where the inner maximization aims to find physical transformations that maximize the cross-entropy loss, while the outer minimization aims to optimize the adversarial example with the worst-case transformations.

#### 3.3.1 Bayesian optimization

To solve problem (6), we search for the worst-case transformations one by one. Given an initialized transformation, we perform gradient-based optimization to update the transformation parameters (e.g., angles for rotations). However, randomly selecting initialized transformations is ineffective since the random initialization may drop into regions of weak transformations, which limits the exploration of the space of all transformations. To address this issue, we propose to adopt the Bayesian optimization [17, 54] to better break the dilemma between exploration and exploitation to find more appropriate initialized transformations.

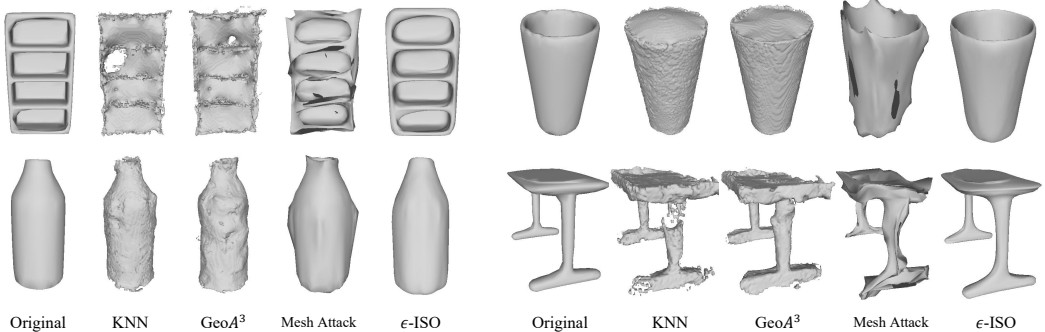

| Original | KNN | $GeoA^3$ | Mesh Attack | $\epsilon$-ISO | Original | KNN | $GeoA^3$ | Mesh Attack | $\epsilon$-ISO |

Figure 3: Adversarial objects obtained by KNN attack, $GeoA^3$ attack, Mesh Attack, and our $\epsilon$-ISO attack against the PointNet model. The KNN and $GeoA^3$ attack produce adversarial samples with dense irregular deformations. The Mesh Attack produces distortions and self-intersecting surfaces (i.e., black surfaces). None of them guarantees naturalness, while $\epsilon$-ISO attack is more natural.

Bayesian optimization is an efficient method for solving global optimization problems consisting of two key components: a surrogate model, such as a Gaussian process (GP) [51] or a Bayesian neural network (BNN) [32], which models the unknown objective; and an acquisition function $\alpha(\cdot)$, which is maximized by balancing exploitation and exploration to recommend the next query location. We choose GP as a surrogate model, which provides a Bayesian posterior distribution to describe the unknown function $\mathcal{L}_{ce}\left(t(S(\mathcal{M}_{adv})), y^*\right)$. We use the expected improvement (EI) [29, 47] acquisition function, which measures the expected improvement of each point with respect to the current best value. Then, the unknown function is sampled by maximizing the acquisition function to better explore the space of all transformations by selecting the next query point in the region where the prediction is high and the model is very uncertain.

As shown in the overall algorithm (in Appendix B), we update the Bayesian posterior distribution of the objective $f$ using the observations obtained from the previous iterations in the gradient descent process. Then, we maximize the EI acquisition function $\alpha_{EI}\left(t; D_n\right)$ to find the initial transformation. From the initialization, we optimize the transformation parameters by gradient-based method to solve problem (6). This process is repeated for the number of transformations in the MaxOT algorithm.

# 4 Experiments

## 4.1 Experimental setup

**Dataset.** We use the ModelNet40 [71] dataset in our experiments. This dataset contains 12,311 CAD models with 40 common object semantic categories in the real world. We use the official split [48, 49], where 9,843 examples are used for training and the remaining 2,468 examples are used for testing. We follow [88] and get closed manifolds. For adversarial attacks, we follow [69] and take all the instances of the 40 categories that are well classified in the ModelNet40 testing set.

**Victim models.** Following [69, 88, 40], we select three commonly used point cloud classification networks as the victim models, i.e., PointNet [48], PointNet++ [49], and DGCNN [68].

**Evaluation metrics.** To quantitatively evaluate the effectiveness of our proposed $\epsilon$-ISO attack, we measure the attack success rate (ASR). Besides, to measure the imperceptibility of different attack methods, we use the Chamfer distance $\mathcal{D}_c$ [15] and Gaussian curvature distance $\mathcal{D}_g$ (i.e., Eq. (3)) as evaluation metrics. We also report the attack success rates on several existing defenses [79, 95, 72] and under the physical world to further validate the superiority of our $\epsilon$-ISO attack.

**Implementation details.** We use Adam [31] to optimize the objective of our proposed $\epsilon$-ISO attack. We use a fixed learning schedule of 250 iterations, where the learning rate and momentum are respectively set as 0.01 and 0.9. We assign the weighting parameters $\lambda_1 = 1.0$, $\lambda_2 = 0.2$ and $\lambda_3 = 0.8$. The balancing parameter $\beta$ is initialized as 1,500 and automatically adjusted by conducting 10 runs of binary search following [4]. We uniformly sample 1,024 points from the adversarial mesh.

Table 1: Quantitative results of KNN attack, $GeoA^3$ attack, Mesh Attack, and our proposed $\epsilon$-ISO attack against different models. Our proposed $\epsilon$-ISO attack outperforms all existing methods in terms of attack success rate (ASR) and imperceptibility. We adopt the Chamfer distance $\mathcal{D}_c$ [15] and Gaussian curvature distance $\mathcal{D}_g$ as the evaluation metrics.

| Model | PointNet | | | PointNet++ | | | DGCNN | | |
|---|---|---|---|---|---|---|---|---|---|
| | ASR | $\mathcal{D}_c$ | $\mathcal{D}_g$ | ASR | $\mathcal{D}_c$ | $\mathcal{D}_g$ | ASR | $\mathcal{D}_c$ | $\mathcal{D}_g$ |
| KNN | 14.78% | 0.0034 | 0.0096 | 6.24% | **0.0027** | 0.0122 | 4.17% | 0.0036 | 0.0125 |
| $GeoA^3$ | 19.65% | 0.0066 | 0.0037 | 11.20% | 0.0192 | 0.0031 | 8.24% | 0.0172 | 0.0042 |
| Mesh Attack | 88.10% | 0.0054 | 0.0048 | 94.79% | 0.0045 | 0.0046 | 66.79% | 0.0051 | 0.0055 |
| $\epsilon$-ISO | **98.45%** | **0.0031** | **0.0004** | **99.58%** | 0.0040 | **0.0004** | **84.16%** | **0.0032** | **0.0009** |

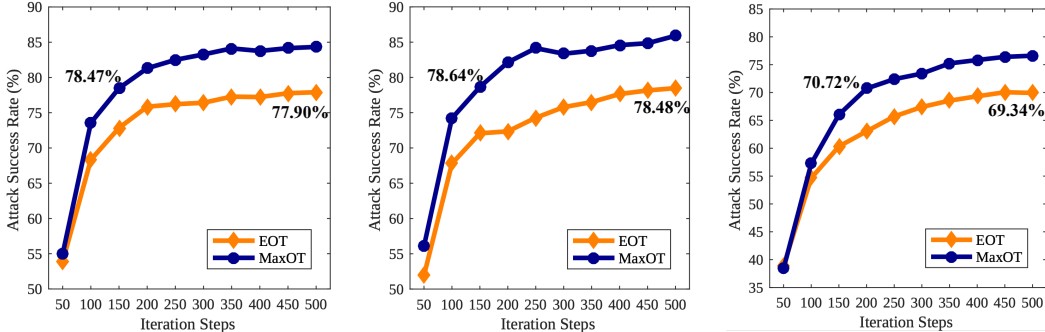

Figure 4: The attack success rate w.r.t number of iterations curves of EOT and MaxOT against PointNet, PointNet++, and DGCNN. Our proposed MaxOT algorithm can lead to a higher attack success rate with a small number of iterations than that of EOT with a large number of iterations.

## 4.2 Pseudo physical attack

In this section, we compare our proposed $\epsilon$-ISO attack with existing methods [63, 69, 88] that generate adversarial objects under the setting of white-box targeted attack, including the KNN attack [63], $GeoA^3$ attack [69], and Mesh Attack [88]. In Table 1, we compare different methods under the measures of attack success rate (ASR), Chamfer distance $\mathcal{D}_c$ [15] and Gaussian curvature distance $\mathcal{D}_g$. Our proposed $\epsilon$-ISO attack outperforms all existing methods in terms of attack success rate and is almost optimal in terms of Chamfer distance $\mathcal{D}_c$ and Gaussian curvature distance $\mathcal{D}_g$, the evaluation metrics for measuring imperceptibility. These comparisons confirm the effectiveness of our proposed regularization term based on Gaussian curvature, to simultaneously achieve the strongest adversarial attack and the most natural perturbations.

Fig. 3 shows the adversarial objects generated by different methods. The KNN attack and $GeoA^3$ attack produce dense irregular deformations on the mesh surface, losing smoothness and regularity. The Mesh Attack creates severe deformations and spurs, which are easily perceived by humans visually. In addition, the severe irregularities and distortions generated by the Mesh Attack can lead to self-intersecting phenomena in the mesh of 3D objects, i.e., the black surfaces in Fig. 3. These self-intersecting surfaces can prohibit 3D-printing the adversarial objects in the physical world. By considering geometric features of the object instead of using metrics in the Euclidean space, our $\epsilon$-ISO attack produces adversarial samples without dense deformations, bursts, distortions, and self-intersection phenomena, while are stealthy and natural.

## 4.3 Robustness under physical transformation

To further verify the superior robustness of our proposed MaxOT algorithm under transformations in the physical world, we compare MaxOT to EOT. In Table 2, we compare the different methods under rotations to calculate the attack success rate, Chamfer distance $\mathcal{D}_c$ and Gaussian curvature distance $\mathcal{D}_g$. Our proposed MaxOT algorithm outperforms the EOT algorithm in terms of attack success rate. This comparison confirms our insight that the resistance to the most harmful physical transformations is better than random ones.

Table 2: Quantitative results of attacking different models using EOT and MaxOT. Our proposed MaxOT algorithm outperforms the EOT algorithm in terms of attack success rate.

| Model | PointNet | | | PointNet++ | | | DGCNN | | |
|---|---|---|---|---|---|---|---|---|---|
| | ASR | $\mathcal{D}_c$ | $\mathcal{D}_g$ | ASR | $\mathcal{D}_c$ | $\mathcal{D}_g$ | ASR | $\mathcal{D}_c$ | $\mathcal{D}_g$ |
| EOT | 76.20% | 0.0074 | 0.0009 | 74.28% | 0.0094 | 0.0007 | 65.72% | 0.0068 | 0.0041 |
| MaxOT | **82.50%** | 0.0074 | 0.0009 | **84.14%** | 0.0094 | 0.0006 | **72.40%** | 0.0067 | 0.0039 |

Table 3: The attack success rate of various methods against PointNet under different defense methods. Our proposed $\epsilon$-ISO attack performs better under stronger defenses, and the attack success rate under DUP-Net and IF-Defence is higher than all other attacks.

| | no defence | SRS | SOR | DUP-Net | IF-Defence |
|---|---|---|---|---|---|
| KNN | 32.15% | 24.34% | 13.36% | 11.25% | 7.46% |
| $GeoA^3$ | 40.87% | 3.34% | 33.22% | 11.97% | 1.04% |
| Mesh Attack | 93.39% | **87.38%** | **89.93%** | 78.28% | 49.06% |
| $\epsilon$-ISO | **99.81%** | 79.85% | 85.42% | **78.55%** | **60.51%** |

Table 4: The success rate of attacking the PointNet model in the physical world using the EOT algorithm and our proposed MaxOT algorithm.

| | PointNet |
|---|---|
| EOT | 72.36% |
| MaxOT | **80.12%** |

In Fig. 4, we give the success rate w.r.t. iterations curves for EOT and MaxOT of different victim models. Our proposed MaxOT algorithm is always ahead of the EOT algorithm at different number of iterations. Moreover, we note that our proposed MaxOT algorithm can lead to higher attack success rate with a small number of iterations than that of EOT with a large number of iterations. Therefore, our proposed MaxOT algorithm is more efficient.

## 4.4 Performance under defense

We further verify the effectiveness of our $\epsilon$-ISO attack under the existing defense methods. To evaluate the adversarial robustness of various attacks, we use the following defense methods: Simple Random Sampling (SRS) [79], Statistical Outlier Removal (SOR) [95], DUP-Net defense [95], and IF-Defense [72]. We give the attack success rate of various attacks under each defense method in Table 3. Under simple defenses such as SRS and SOR, the attack success rate of Mesh Attack is higher than our $\epsilon$-ISO attack.

However, under more advanced and effective defenses such as IF-Defence, the attack success rate of our $\epsilon$-ISO attack is higher than all other attacks. This is because KNN attack and $GeoA^3$ attack generate adversarial objects through dense local deformation, generating outliers that are easily detected by the defense methods. The Mesh Attack generates adversarial objects through large anomalous deformation, resulting in fewer local outliers that have advantages under simple statistical defenses such as SRS and SOR, but cannot effectively attack the IF-Defence with better defense performance. Our $\epsilon$-ISO attack produces adversarial samples without local outliers or anomalous deformations, with better attack performance, especially under IF-Defence.

## 4.5 Physical attack

We randomly select 100 instances in Table 2, 50 of which are generated by the MaxOT algorithm and the other 50 are the corresponding instances generated by the EOT algorithm. The selected meshes are printed by the Stratsys J850 Prime 3D printer and scanned by the EinScan-SE 3D scanner. The attack success rate in the physical world is shown in Table 4, where our algorithm almost completely maintains the effect of the pseudo physical world attack. None of the previous algorithms could maintain the pseudo physical world attack well in the physical world because the dense deformation and self-intersection phenomena would interfere with the scanning and prevent the scanner from correctly scanning the surface of the adversarial samples generated by their algorithms. In contrast, the surface of the adversarial sample produced by our $\epsilon$-ISO attack is smooth and natural and can be scanned correctly by the scanner, preserving the adversarial effect. In addition, the improvement on the attack success rate of our proposed MaxOT algorithm against the EOT algorithm is also reflected in the physical world. Our proposed MaxOT algorithm is necessary to improve the robustness of the adversarial sample under the transformation of the physical world. The adversarial meshes, 3D printed physical meshes, and scanned point clouds are shown in Fig. 5.

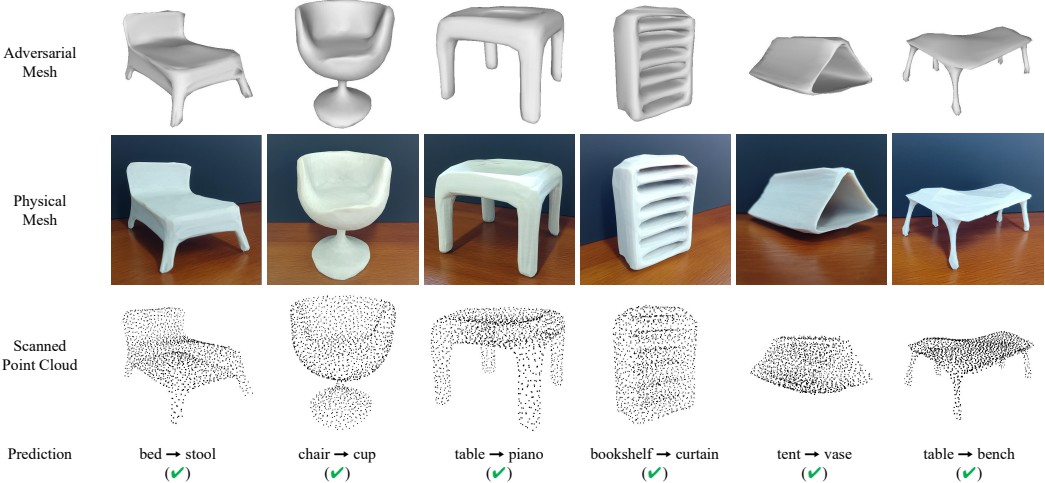

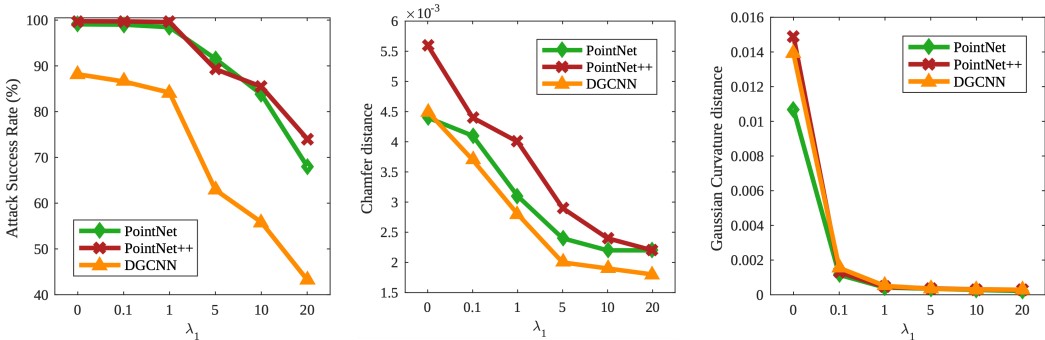

Figure 5: Visualization of the $\epsilon$-ISO attack in the physical world. The adversarial mesh is randomly selected from Table 2. After 3D-printing and scanning, the point cloud obtained from the scanning is used to attack the PointNet. A green check mark indicates a successful attack. The adversarial samples we generated are natural and stealthy, and because of their smoothness and naturalness, can be scanned correctly by the 3D-scanner, maintaining the adversarial effect in the physical world.

Figure 6: The effects of the penalty parameter $\lambda_1$ of the Gaussian curvature consistency regularization term. When $\lambda_1$ is large, the attack success rate decreases rapidly. When $\lambda_1$ is small, the naturalness of the adversarial objects is worse. The default $\lambda_1 = 1$ gives the best result for balancing the attack success rate and stealthiness.

## 4.6   Ablation studies

To investigate the effect of the penalty parameter $\lambda_1$ of the Gaussian curvature consistency regularization term on the attack success rate and imperceptibility metrics, we adjust the value of $\lambda_1$ to perform quantitative and qualitative experiments. The plots in Fig. 6 show that when $\lambda_1$ is tuned high, the attack success rate decreases rapidly. When $\lambda_1$ is adjusted lower, the values of the evaluation metrics increase and deteriorate rapidly. The qualitative visualization results in Appendix C demonstrate the irregularity of the 3D objects as $\lambda_1$ is turned down. Our default value of $\lambda_1 = 1$ gives the best result for balancing the attack success rate and stealthiness. To investigate the effectiveness of Bayesian optimization to find the initial transformations, we test the attack success rate for randomly selecting the initial transformation in the MaxOT algorithm. As shown in Appendix C, the improvement of using Bayesian optimization over random initialization in the MaxOT algorithm is significant. This confirms the importance of Bayesian optimization methods in our approach. More ablation studies can be found in Appendix C.

## 5   Conclusion

In this paper, we propose a novel $\epsilon$-isometric ($\epsilon$-ISO) attack method to generate natural and robust 3D adversarial examples in the physical world. We improve the naturalness of 3D adversarial examples by constraining it to be $\epsilon$-isometric to the original one. Through a theoretical analysis, we adopt

the Gaussian curvature as the surrogate metric. We further propose a maxima over transformation (MaxOT) algorithm armed with Bayesian optimization that actively searches for the most harmful transformations rather than random ones to make the generated adversarial example more robust in the physical world. Extensive Experiments on typical point cloud recognition models validate the effectiveness of our approach in terms of attack success rate and naturalness compared to the state-of-the-art methods. A potential negative societal impact of our work in that the malicious adversaries may adopt our method to generate 3D adversarial objects in the physical world, which can cause severe security/safety consequences for real-world applications. Thus it is imperative to develop more robust 3D recognition models, which we leave to future work.

## Acknowledgement

Yibo Miao and Xiao-Shan Gao were supported by NKRDP grant No.2018YFA0704705 and NSFC grant No.12288201. Yinpeng Dong and Jun Zhu were supported by the National Key Research and Development Program of China (2020AAA0106000, 2020AAA0104304, 2020AAA0106302), NSFC Projects (Nos. 62061136001, 62076145, 62076147, U19B2034, U1811461, U19A2081, 61972224), Beijing NSF Project (No. JQ19016), BNRist (BNR2022RC01006), Tsinghua Institute for Guo Qiang, and the High Performance Computing Center, Tsinghua University. Y. Dong was also supported by the China National Postdoctoral Program for Innovative Talents and Shuimu Tsinghua Scholar Program. J. Zhu was also supported by the XPlorer Prize.

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
