# OpenReview forum: "Isometric 3D Adversarial Examples in the Physical World"
_NeurIPS.cc/2022/Conference — NeurIPS 2022 Accept_

### Official Review · Reviewer_WZCq · 2022-07-10

**Rating:** 5
**Confidence:** 4
**Soundness:** 3 good
**Presentation:** 3 good
**Contribution:** 2 fair

**Summary:**

This paper presents $\epsilon$-ISO  attack method to generate natural and robust 3D adversarial examples in the physical world. The authors present a new metric or definition on the naturalness of the 3D adversarial example. They also provide theoretical proof to demonstrate that Gaussian curvature could be used as a proxy for $\epsilon$-isometric. MaxOT is proposed as a method combined with Bayesian to optimize the 3D adversarial examples.

**Questions:**

See the first point in the weakness section.

**Limitations:**

The authors should discuss the potential societal impact of the proposed attack.

**Strengths And Weaknesses:**

Strengths:
1. The paper contains both formal and empirical results to demonstrate the effectiveness of the proposed method.
2. The organization of the paper is good, and the content is rich.
3. The authors leverage extensive evaluation to show that the proposed attack is effective and realistic in the real world. The experiments with 3D printed mesh are convincing.

Weakness:
1. Although the authors successfully demonstrate the physical attack on 3D point cloud recognition task, it is still questionable how these attacks will have real-world impacts. Real-world application of point clouds mostly include object detection and segmentation. Delving into the recognition task and showing naturalness seem to be off the correct direction in this area.
2. I like the organization of this paper, but the content is too crowded and sometimes it is hard to follow every detail.

---

> ### Author Response · Authors · 2022-08-02
> **Thank you for the valuable review**
>
> Thank you for appreciating our contributions as well as providing the valuable feedback. We have uploaded a revision of our paper. Below we address the detailed comments, and hope that you can find our response satisfactory.
>
> ***Question 1: Although the authors successfully demonstrate the physical attack on 3D point cloud recognition task, it is still questionable how these attacks will have real-world impacts. Real-world application of point clouds mostly include object detection and segmentation.***
>
> First, point cloud recognition is the most fundamental task of point clouds. There are many typical models developed for point cloud recognition, such as PointNet and DGCNN, which are used as backbones for point cloud detection and segmentation models. Therefore, studying adversarial attacks on these typical models in the fundamental point cloud recognition task is meaningful to understand the vulnerabilities of these backbone models.
>
> Second, although our proposed $\epsilon$-ISO and MaxOT are only applied to the point cloud recognition task, the techniques can be also extended to other tasks, such as point cloud detection and segmentation.
> The proposed $\epsilon$-ISO attack constrains the adversarial example to be $\epsilon$-isometric to the original one, and it could be generally applicable to other tasks to maintain the geometric properties of the 3D objects.
> The proposed MaxOT algorithm actively searches for the most harmful transformations rather than random ones to make the generated adversarial example more robust in the physical world. This intuition is also general for other tasks.
> So our algorithms can be easily extended to other tasks to improve the naturalness and robustness of 3D adversarial objects.
> We leave the extension of our methods to point cloud detection and segmentation in future work.
>
> ***Question 2: I like the organization of this paper, but the content is too crowded and sometimes it is hard to follow every detail.***
>
> Thanks for the suggestion. In the revision, we rewrite some parts of this paper (e.g., Section 3.3.1) to make the content easy to follow. We will further improve the writing in the final.
>
> ***Question 3: The authors should discuss the potential societal impact of the proposed attack.***
>
> Thanks for the suggestion. In fact, we have already discussed the potential societal impact of the proposed attack in Section 5.

---

### Official Review · Reviewer_FQV5 · 2022-07-11

**Rating:** 8
**Confidence:** 5
**Soundness:** 4 excellent
**Presentation:** 3 good
**Contribution:** 4 excellent

**Summary:**

This paper proposes a novel $\epsilon$-ISO attack algorithm to generate natural and robust 3D adversarial samples. It is motivated by the fact that the distance metric on the Euclidean space does not consider geometric features and cannot accurately measure the similarity between 3D objects. The proposed method constrains the adversarial sample to be $\epsilon$-isometric to the original sample to ensure the geometric similarity and adopts the Gaussian curvature as the regularization. The theoretical analysis is provided to uncover the connection between isometry and Gaussian curvature. The authors further propose MaxOT to improve the robustness of 3D adversarial samples in the physical world. MaxOT is reasonable by actively searching for the most harmful transformations during optimization. The experiments validate the naturalness and effectiveness of the proposed 3D attack.

**Questions:**

### Other comments:
I was inspired by the authors’ discussion of metrics on Non-euclidean space. As analyzed by the authors, the distance metric on Euclidean space is not a perfect representation of the similarity between 3D objects because it does not take geometric features into account. While reading, I kept thinking if the Fisher information metric on statistical manifold mapped from topological manifold [1] would be an interesting consideration.

[1] F. Li et al, "Toward a unified framework for point set registration," ICRA 2021


Overall, I think the strengths of this paper overweigh its weaknesses. The contributions of this paper could be significant for the venue. I’d like to see my concerns addressed by the authors and consider increasing the rating.

### Question:
- In fig4, the title of the y-axis should be the attack success rate.
- Line 16 of the appendix: The authors should further clarify the relationship between Gaussian curvature and the first fundamental form in the proof in the appendix to ensure that it can be clearly understood.
- Line 142: "bending degree" should be "the bending degree".
- Line 148: "bending degree" should be "the bending degree".
- Line 149: "Definition" should be "Definitions".
- Line 262: "higher" should be "a higher".
- table 3: "DUP-Net" should be "DUP-Net defense".
- Line 281: "selecte" should be "select".
- In fig6(a), the title of the y-axis should be the attack success rate.

**Ethics Review Area:**

["I don’t know"]

**Limitations:**

/na

**Strengths And Weaknesses:**

Strengths:
- First, the paper is easy to read. The algorithm and experiments are well organized and explained. The experiments are thorough to prove the effectiveness.
- Overall, I find the proposed approach novel. To the best of my knowledge, previous 3D adversarial attacks only considered the distance metric on Euclidean space, i.e. Hausdorff distance, Chamfer distance, etc., whereas this is the first work to consider the geometric properties on Non-euclidean space of 3D objects. This paper introduces $\epsilon$-isometric mapping to ensure the geometric similarity between the adversarial sample and the original one. It further connects $\epsilon$-isometry to the Gaussian curvature, which can be computed and optimized in generating 3D adversarial samples. Furthermore, the MaxOT algorithm is reasonable to improve the robustness of 3D adversarial samples.
- The theoretical analysis makes this paper more solid and convincing.
- The paper has conducted thorough evaluation comparisons between the proposed attack and other attacks. The performance improvement is obvious. The 3D adversarial objects look more natural according to the visualization. The effectiveness measured by the attack success rate is also validated.

Weaknesses:
- Table 2 and Figure 4 demonstrate that the proposed MaxOT outperforms EOT in terms of attack success rate and efficiency in the physical world. Is it due to the 3D objects generated by MaxOT are more perceptible?
- To further verify the effectiveness of the proposed method, the authors are encouraged to conduct experiments of black box attacks.

---

> ### Author Response · Authors · 2022-08-02
> **Thank you for the valuable review**
>
> Thank you for appreciating our new contributions as well as providing the valuable feedback. We have uploaded a revision of our paper. Below we address the detailed comments.
>
> ***Question 1: Are 3D objects generated by MaxOT more perceptible?***
>
> As shown in Table 2, MaxOT and EOT achieve very similar results in naturalness of the generated adversarial objects.
> Besides, as shown in Figure C.3 in Appendix C.4, the qualitative visualization results of MaxOT and EOT also show the same degree of naturalness and imperceptibility of the adversarial objects.
> Our proposed MaxOT algorithm has better results because it actively searches for the most harmful physical transformations, rather than inducing more perceptible perturbations.
>
> ***Question 2: The authors are encouraged to conduct experiments of black box attacks.***
>
> Thanks for the valuable suggestion. We further conduct experiments on the transfer-based attacks. We generate 3D adversarial examples against one white-box model and evaluate the black-box attack success rates on the other black-box victim models. The results are shown below.
>
> | White-box Model | Attacks | | Black-box Model | |
> | :-----: | :----: | :----: | :-----: | :----: |
> | | | PointNet | PointNet++ | DGCNN |
> |  | KNN | - | 11.1\% | 10.7\% |
> |PointNet | $GeoA^3$ | - | 11.5\% | 2.5\% |
> || $\epsilon$-ISO | - | 47.6\% | 35.8\% |
> | | KNN | 6.4\% | - | 7.9\% |
> |PointNet++ | $GeoA^3$ | 9.4\% | - | 19.7\% |
> |  | $\epsilon$-ISO | 32.9\% | - | 51.2\% |
> |  | KNN | 7.2\% | 32.2\% | - |
> | DGCNN | $GeoA^3$ | 12.4\% | 24.2\% | - |
> |  | $\epsilon$-ISO | 55.4\% | 62.7\% | - |
>
> It can be seen that our $\epsilon$-ISO attack has much higher success rates than the baselines. This is because our $\epsilon$-ISO retains the geometric properties of the 3D objects well, without local outliers or anomalous deformations. Thus the crafted adversarial examples more transferable across different models. In the revision, we add the results in Appendix C.6.
>
> ***Question 3: If the Fisher information metric on statistical manifold mapped from topological manifold would be an interesting consideration.***
>
> Thanks for the valuable suggestion.
> The idea of giving a Riemannian metric to statistical manifold to study similarity is in some sense elegant. The 3D data can be represented as samples drawn from a statistical distribution and mapped to the underlying statistical manifold. After giving a Riemannian metric to the statistical manifold, a similarity metric can be determined based on the minimal geodesic curve [1].
> This idea may be useful in the field of 3D adversarial attacks and defenses.
> However, this idea largely boils down to the standard assumption that a point cloud is a set of samples from an underlying surface, perturbed by random noise. So, the view as a distribution is probably not suitable for data with some sharp local features.
> We do agree with the reviewer that this is a promising direction for measuring similarity between 3D point clouds.
> We leave it to our future work.
>
> ***Question 4: Line 16 of the appendix: The authors should further clarify the relationship between Gaussian curvature and the first fundamental form in the proof in the appendix to ensure that it can be clearly understood.***
>
> Thanks for the suggestion.
> The first fundamental form is the expression of how the surface $S$ inherits the natural inner product of $\mathbb{R}^{3}$. Geometrically, the first fundamental form allows us to make measurements on the surface (e.g., lengths of curves, angles of tangent vectors, areas of regions) without referring back to the ambient space $\mathbb{R}^{3}$ where the surface lies. The second fundamental form describes the shape of the surface in the ambient space $\mathbb{R}^{3}$. The Gaussian curvature can be defined by the coefficients of the first fundamental form and the coefficients of the second fundamental form. The Gauss formula and the Mainardi-Codazzi equations reveal the relations between the first and second fundamental forms of a surface. Gauss formula expresses the Gaussian curvature as a function of the coefficients of the first fundamental form and its derivatives, i.e., Eq. (A.1), which is also known as Gauss’ Theorema Egregium [2]. In the revision, we clarify this in Remark 2 in Appendix A.
>
> ***Question 5: Minor writing issues.***
>
> Thanks for pointing out these issues. In the revision, we fix these issues and improve the writing. We will further polish our paper in the final.
>
> Reference:
>
> [1] Petersen, Peter. Riemannian geometry. Vol. 171. New York: Springer, 2006.
>
> [2] Gauss, K. F., General Investigations of Curved Surfaces, Raven Press, New York, 1965.

---

> > ### Comment · Reviewer_FQV5 · 2022-08-03
> > **Response to authors**
> >
> > I think the author answered most of my questions, so I'll raise my score to 8

---

> > > ### Author Response · Authors · 2022-08-09
> > > **Thank you for the update!**
> > >
> > > Thank you very much for the update and increasing the score!
> > >
> > > Best,
> > > Authors

---

### Official Review · Reviewer_mBRv · 2022-07-12

**Rating:** 6
**Confidence:** 4
**Soundness:** 4 excellent
**Presentation:** 3 good
**Contribution:** 4 excellent

**Summary:**

The paper proposes a white-box adversarial attack on 3D point cloud recognition models such that the corresponding attacks are robust, natural, and reproducible in the physical world. For naturalness, they propose a novel framework based on maintaining $\epsilon$-isometry between the original and adversarial point cloud model. To boost robustness and ensure reproducibility, they develop a new framework combining maxima over transformation with bayesian optimization. Experimental evaluation against existing attacks on state-of-the-art models shows that the proposed approach achieves higher attack success rate while remaining imperceptible. The authors further show reproducibility in the real world.

**Questions:**

* In Section 3.1, $\mathcal{X}$ is undefined. I assume it's the set of all meshes or point clouds? Line 101, how is $\Delta$ measured? Is the sampling function $\mathcal{S}$ fixed for a given mesh $\mathcal{M}$? Can you provide a mathematical definition of the length function "s" used in Definition 1? How do you get the equality on line 145?

* Line 132, "Therefore, we introduce the definition of the Gaussian curvature [11] of the surface", is it your definition or taken from literature? Please be precise with the wording.

* In Definition 1, what is the range of values of n? For arbitrarily n, it looks like the difference in the lengths of curves on $S$ and $\hat{S}$ can be quite large?

* In Definition 2, can you explain how the parameters $(u,v)$ relate to the meshes?

*  Is Theorem 1 your contribution or adapted from prior work?

* Can you compare the runtime of the different baselines and your method?

* "The Mesh Attack generates adversarial objects through large anomalous deformation, resulting in fewer local outliers that have advantages under simple statistical defenses such as SRS and SOR, but cannot effectively attack the DUP-Net defense and IF-Defence with better defense performance. " While there is a significant improvement over the mesh attack under the IF-defence, improvement under the DUP-Net defense is marginal (<0.5%), so it would be misleading to claim that mesh attack is ineffective and your attack is effective in this case.

* Can your MaxOT + Bayesian optimization provide any formal guarantees on the robustness of your examples (e.g., as in https://arxiv.org/pdf/2007.12133.pdf)? Further, does your robustness enable you to be more effective against certified defenses based on randomized smoothing (e.g., https://arxiv.org/abs/2103.03046)?

* Why did you not analyze the effects of the hyperparameters $\lambda_2$ and $\lambda_3$?

* How do $\mathcal{D}_c$ and $\mathcal{D}_g$ relate to $\epsilon$ in your $\epsilon$-ISO attacks?

* I observed you only consider rotations when considering robustness in experiments. Any particular reason for that? Why not consider the composition of translation, rotation, other transformations, etc.?

**Limitations:**

The authors have adequately addressed the limitations and potential negative impacts of their work.

**Strengths And Weaknesses:**

Strengths:
+ The paper considers a novel setting where the author's goal is to generate robust and natural adversarial examples for NNs working on  3-D point clouds that can also be realized in the physical world.

+ The technical contributions are novel and tailored for creating physical, natural, and robust adversarial examples in the 3-D point cloud setting.

+ Strong experimental results showing the value of the contributions including reproducibility in the real world.

Weaknesses:
- The technical details in Section 3.1 are not rigorous enough which makes the corresponding text hard to follow. Further, while the high-level idea in 3.3.1 is clear, the text is too dense. I suggest the authors improve their writing at these places for making their technical contributions more accessible.

-  While input-specific white-box attacks show the non-robustness of NNs, they are usually not realistic. The authors do not mention cases where an attacker can gain white-box access to the point cloud model and how it knows what inputs will be processed by the model in advance and whether it can create physical examples fast enough before the input is processed by the model. For example, in the case of a self-driving car, the model will continuously process new inputs. Is your attack feasible in such settings? Therefore I am not sure if the proposed method reveals any vulnerability that can be exploited by a real-world adversary.

---

> ### Author Response · Authors · 2022-08-02
> **Thank you for the valuable review (Part 3/3)**
>
> ***Question 12: How do $D_c$ and $D_g$ relate to $\epsilon$ in your $\epsilon$-ISO attacks?***
>
> $D_g$ is the Gaussian curvature distance defined in Eq. (3). So $D_g$ is less than $\epsilon^2$ given Theorem 1 and Eq. (3).
> $D_c$ is the Chamfer distance, which measures the distance between the two point sets by finding for each point $v^{\prime}$ in $\mathcal{V}_{adv}$ the closest point $v$ in $\mathcal{V}$ and averaging all the distances. $D_c$ is not designed from a geometric point of view, so it is not related to $\epsilon$.
>
> ***Question 13: I observed you only consider rotations when considering robustness in experiments.***
>
> We note that some transformations, such as translations and affine projections, can be easily restored by the input normalization and are thus meaningless.
> Some other transformations, such as distortions, will not appear on the rigid 3D objects in the physical world. Therefore, we only consider 3D rotations, which are more common in the physical world.
> In addition, the success rate of the attack in the real-world experiments is already close to the success rate of the attack in the digital-world experiments when we consider only the rotation transformations. This means that we do not need to consider other transformations.
>
> Reference:
>
> [1] Eykholt et al. Robust Physical-World Attacks on Deep Learning Models. CVPR 2018.
>
> [2] Ravi et al. Accelerating 3D Deep Learning with PyTorch3D. SIGGRAPH 2020.

---

> > ### Comment · Reviewer_mBRv · 2022-08-09
> > **Post Rebuttal Comments**
> >
> > Dear Authors,
> >
> > Thanks for your response, it addressed my concerns. I will update my rating to reflect this.

---

> > > ### Author Response · Authors · 2022-08-09
> > > **A reminder for your rating**
> > >
> > > Appreciate for spending considerable time on our paper. We would also like to remind you that you have not updated your rating in the OpenReview system. Thanks!

---

> ### Author Response · Authors · 2022-08-02
> **Thank you for the valuable review (Part 2/3)**
>
> ***Question 4: Is the definition of the Gaussian curvature your definition or taken from literature?***
>
> This definition is taken from [11]. In the revision, we modify this sentence to "Therefore, we give the definition of the Gaussian curvature of the surface from [11]".
>
> ***Question 5: In Definition 1, what is the range of values of $n$?***
>
> The range of values of $n$ depends on the density of the point cloud. If the point cloud tends to be infinitely dense, then $n$ tends to be 0. In the experiments, we randomly sampled 1024 points and the range of values of $n$ is small enough that the difference in the lengths of curves on $S$ and $\tilde { S }$ can not be quite large.
>
> ***Question 6: In Definition 2, can you explain how the parameters $(u, v)$ relate to the meshes?***
>
> A mesh can be seen as a surface in $\mathbb{R}^3$. Mathematically, a surface $S$ in $\mathbb{R}^3$ can be represented by a function  $\boldsymbol{r}(u, v)=[x(u, v), y(u, v), z(u, v)]$, where $(u, v) \in \mathbb{R}^2$. $(u, v)$ is called a parametrization of the surface.
>
> ***Question 7: Is Theorem 1 your contribution or adapted from prior work?***
>
> Theorem 1 is our contribution. In the proof in Appendix A, we apply the existing results Lemma 1, Lemma 2, and Gauss formula to prove Theorem 1.
>
> ***Question 8: Can you compare the runtime of the different baselines and your method?***
>
> Thanks for the suggestion. We further calculate the average runtime of each attack to generate 3D adversarial objects, which are measured on the same RTX 3080Ti GPU. The results are shown below.
>
> ||PointNet|PointNet++|DGCNN|
> |:-----|:-----:|:----:|:----:|
> |KNN|66.2s|75.3s|67.8s|
> |$GeoA^3$|154.7s|169.0s|158.2s|
> |Mesh Attack|0.3s|3.1s|0.9s|
> |$\epsilon$-ISO|0.3s|3.2s|0.8s|
>
> The average runtime cost of KNN and $GeoA^3$ is much higher than Mesh Attack and $\epsilon$-ISO. This is because KNN and $GeoA^3$ need to reconstruct adversarial meshes, which consumes a lot of time. In the revision, we add the results in Appendix C.7.
>
> ***Question 9: The improvement under the DUP-Net defense is marginal (<0.5\%), so it would be misleading to claim that mesh attack is ineffective and your attack is effective in this case.***
>
> Thanks for pointing this out. In the revision, we revise our argument to make it not misleading.
>
> ***Question 10: Can your MaxOT + Bayesian optimization provide any formal guarantees on the robustness of your examples? Does your robustness enable you to be more effective against certified defenses based on randomized smoothing?***
>
> Thanks for pointing out the related work. Our MaxOT + Bayesian optimization can only boost the empirical robustness of adversarial examples in the physical world. We will consider the formal guarantees of it in future work.
> For the certified defense, we found that there is no open source code of this work, thus we cannot conduct experiments on it for now. We will further study the performance of our method against certified defenses in future work.
>
> ***Question 11: Why did you not analyze the effects of the hyperparameters $\lambda_{2}$ and $\lambda_{3}$?***
>
> Thanks for the suggestion. We further conduct an ablation experiment on $\lambda_{2}$ and $\lambda_{3}$. The results are shown in Figure C.4 in Appendix C.5 and below.
>
> |Model|$\lambda_{2}=0$|$\lambda_{2}=0.02$|$\lambda_{2}=0.2$|$\lambda_{2}=1$|$\lambda_{2}=2$|$\lambda_{2}=4$|
> |:-----|:-----:|:----:|:----:|:-----:|:----:|:----:|
> |PointNet|98.72%|98.67%|98.45%|94.22%|78.03%|69.45%|
> |PointNet++|99.69%|99.66%|99.58%|89.25%|79.82%|72.42%|
> |DGCNN|85.13%|84.82%|84.16%|69.76%|60.48%|52.46%|
>
> | Model | $\lambda_{3}=0$ | $\lambda_{3}=0.08$ | $\lambda_{3}=0.8$ | $\lambda_{3}=4$ | $\lambda_{3}=8$ | $\lambda_{3}=16$ |
> | :----- | :-----: | :----: | :----: | :-----: | :----: | :----: |
> | PointNet | 99.02\% | 98.86\% | 98.45\% | 89.29\% | 78.42\% | 63.01\% |
> | PointNet++ | 99.73\% | 99.70\% | 99.58\% | 86.56\% | 80.47\% | 71.96\% |
> | DGCNN| 87.28\% | 86.89\% | 84.16\% | 65.92\% | 56.92\% | 42.35\% |
>
> We observe that when $\lambda_{2}$ or $\lambda_{3}$ is tuned high, the attack success rate decreases rapidly. When $\lambda_{2}$ is adjusted lower, 3D objects show local unevenness and minor self-intersection. When $\lambda_{3}$ is adjusted lower, 3D objects show large areas of self-intersection. Our default value of $\lambda_{2}=0.2$ and $\lambda_{3}=0.8$ give the best result for balancing the attack success rate and stealthiness. In the revision, we add the results in Appendix C.5.

---

> ### Author Response · Authors · 2022-08-02
> **Thank you for the valuable review (Part 1/3)**
>
> Thank you for appreciating our new contributions as well as providing the valuable feedback. We have uploaded a revision of our paper. Below we address the detailed comments, and hope that you can find our response satisfactory.
>
> ***Question 1: Improve the writing for making their technical contributions more accessible.***
>
> Thanks for the suggestion. In the revision, we clarify the technical details in Section 3.1 to make them more rigorous. We also improve the writing in Section 3.3.1 to make the text clearer. We will further revise the paper in the final.
>
> ***Question 2: While input-specific white-box attacks show the non-robustness of NNs, they are usually not realistic.***
>
> ***Question 2.1: The authors do not mention cases where an attacker can gain white-box access to the point cloud model.***
>
> White-box attacks are widely studied in the literature to identify the vulnerabilities of deep learning models. They are commonly used to evaluate the worst-case robustness of the models before they are deployed. Thus studying white-box attacks is highly useful in the field, although they are usually less realistic for real-world applications. Besides, adversarial examples are transferable, which can enable more practical black-box attacks. We further show the black-box attack success rates of our method and baselines below.
>
> |White-box Model|Attacks||Black-box Model||
> |:-----:|:----:|:----:|:-----:|:----:|
> |||PointNet|PointNet++|DGCNN|
> ||KNN|-|11.1%|10.7%|
> |PointNet|$GeoA^3$|-|11.5%|2.5%|
> ||$\epsilon$-ISO|-|47.6%|35.8%|
> ||KNN|6.4%|-|7.9%|
> |PointNet++|$GeoA^3$|9.4%|-|19.7%|
> ||$\epsilon$-ISO|32.9%|-|51.2%|
> ||KNN|7.2%|32.2%|-|
> |DGCNN|$GeoA^3$|12.4%|24.2%|-|
> ||$\epsilon$-ISO|55.4%|62.7%|-|
>
> It can be seen that our $\epsilon$-ISO attack has much higher success rates than the baselines. This is because our $\epsilon$-ISO retains the geometric properties of the 3D objects well, without local outliers or anomalous deformations. Thus the crafted adversarial examples more transferable across different models. In the revision, we add the results in Appendix C.6.
>
> ***Question 2.2: The authors do not mention how an attacker knows what inputs will be processed by the model in advance and whether it can create physical examples fast enough before the input is processed by the model.***
>
> For almost all physical-world attacks, the attacker does not need to create physical adversarial samples in real-time. In general, the attacker first generates adversarial objects and places them in the physical world. When the victim's sensors collect data in the scenario, the target model will incorrectly recognize the generated adversarial samples, which can cause serious security consequences. For example, in the case of self-driving cars, the attacker can generate an adversarial road sign [1] that is placed on the side of the highway. Any self-driving car passing this adversarial road sign at any time will incorrectly recognize this road sign, resulting in bad decisions and potential crashes.
>
> ***Question 3: Writing issues in Section 3.1.***
>
> Thanks for pointing out these issues. We further revise our paper to make it clearer. Below we answer each question.
>
> ***Question 3.1: $\mathcal{X}$ is undefined. I assume it's the set of all meshes or point clouds?***
>
> Yes, $\mathcal{X}$ is the set of all point clouds. In the revision, we clarify this notation.
>
> ***Question 3.2: Line 101, how is $\Delta$ measured?***
>
> Actually, we do not measure the perturbation $\Delta$. To improve the naturalness of the adversarial 3D objects, we make an adversarial example $\epsilon$-isometric to the original one by adopting the Gaussian curvature as a surrogate metric.
> The previous approaches that directly constrain the perturbation $\Delta$ with $L_{p}$ norms fail to maintain the geometric properties. Thus we prefer to constrain the adversarial 3D objects rather than the perturbations.
>
> ***Question 3.3: Is the sampling function $S$ fixed for a given mesh $\mathcal{M}$?***
>
> No, the sampling function $S$ based on Pytorch3D [2] is fully random for a given mesh $\mathcal{M}$.
>
> ***Question 3.4: Can you provide a mathematical definition of the length function "$s$" used in Definition 1?***
>
> Mathematically, a curve $C$ in $\mathbb{R}^{3}$ can be represented by a function $\boldsymbol{r}(t) = [x(t), y(t), z(t)]$, where $t$ in a real number in the interval $(a,b)$. The length of $C$ is by definition $s(C)=\int_{a}^{b}\left|\boldsymbol{r}^{\prime}(t)\right| d t$, where $\left|\boldsymbol{r}^{\prime}(t)\right|=\sqrt{\left(x^{\prime}(t)\right)^{2}+\left(y^{\prime}(t)\right)^{2}+\left(z^{\prime}(t)\right)^{2}}$ is the length of the vector $\boldsymbol{r}^{\prime}(t)$.
> Intuitively, $s(C)$ measures the length of the curve $C$ after straightening it.
>
> ***Question 3.5: How do you get the equality on line 145?***
>
> In the revision, we provide the proof of this equality in Appendix D.

---

> ### Author Response · Authors · 2022-08-09
> **Looking forward to further feedback**
>
> Dear Reviewer mBRv,
>
> Thanks again for appreciating our contributions as well as providing valuable comments. We have carefully addressed them in detail. As the rebuttal is about to close, we hope you may find the response satisfactory (as the other reviewers), and we are happy to address further feedback (if any).
>
> Best,
> Authors

---

### Author Response · Authors · 2022-08-05
**Look forward to further feedback**

Dear reviewers,

We thank you again for the valuable and constructive comments. We are looking forward to hearing from you about any further feedback.

If you find our response satisfactory, we hope you might view this as a sufficient reason to further raise your rating.

If you still have questions about our paper, we are willing to answer them and improve our paper.

Best, Authors

---

### Meta-Review · Area_Chair_Jb8d · 2022-09-09

**Recommendation:** Accept
**Confidence:** Less certain

**Metareview:**

The authors propose a new method for constructing adversarial distortions of 3d objects. The method centers around the a new distance metric between 3D objects which the authors argue is better suited for natural looking perturbations relative to the euclidian metric. Reviewers overall felt the paper was strong, the results convincing and the method new and interesting. The primary concerns were raised by reviewer WZCq, who questioned whether or not the method can be expected to have practical impact. The AC agrees with these concerns, particularly given that most documented examples of attacks on real systems do not involve any notion of subtleness, nor do attackers seem motivated to constrain adversarial inputs to small perturbations of clean inputs [1]. However, given the strong technical contribution of the work and the overall strong reviews, the AC recommends accepting the work but encourages authors to consider revising the text noting that epsilon, (or small, or subtle) perturbations is in no means a strict constraint on the attacker action space in real world settings.

1. Gilmer et. al. "Motivating the rules of the game for adversarial example research", 2018.

**Award:**

No

---

### Decision · Program_Chairs · 2022-09-14

Accept